# Life under quartz: Hypolithic mosses in the Mojave Desert

**Jenna T. B. Ekwealor** [1] *, **Kirsten M. Fisher** [2]

**1** Department of Integrative Biology, University of California, Berkeley, California, United States of America,
**2** Department of Biological Sciences, California State University, Los Angeles, California, United States of America

* jtbe@berkeley.edu

**Data Availability Statement:** Data and analysis code have been deposited into github and made publicly available at: https://github.com/jenna-tb-ekwealor/hypolithic-moss.

**Funding:** This work was supported by the California State University, Los Angeles,

## Abstract

Several species of dryland cyanobacteria are known to occur as hypoliths under semi-translucent rocks. In the Mojave Desert, these organisms find refuge from intense solar radiation under milky quartz where moisture persists for a longer period of time than in adjacent soil surface habitat. Desert mosses, which are extremely desiccation-tolerant, can also occur in these hypolithic spaces, though little is known about this unique moss microhabitat and how species composition compares to that of adjacent soil surface communities. To address this question, we deployed microclimate dataloggers and collected moss samples from under and adjacent to 18 milky quartz rocks (quartz mean center thickness 26 ± 15 mm) in a western high elevation Mojave Desert site. Light transmission through Mojave quartz rocks may be as low as 1.2%, and data from microclimate loggers deployed for five months support the hypothesis that quartz provides thermal buffering and higher relative humidity compared to the soil surface. Of the 53 samples collected from hypolith and surface microhabitats, 68% were *Syntrichia caninervis*, the dominant bryophyte of the Mojave Desert biological soil crust. *Tortula inermis* accounted for 28% of the samples and 4% were *Bryum argenteum*. In a comparison of moss community composition, we found that *S. caninervis* was more likely to be on the soil surface, though it was abundant in both microhabitats, while *T. inermis* was more restricted to hypoliths, perhaps due to protection from temperature extremes. In our study site, the differences between hypolithic and surface microhabitats enable niche partitioning between *T. inermis* and *S. caninervis*, enhancing alpha diversity. This work points to the need to thoroughly consider microhabitats when assessing bryophyte species diversity and modelling species distributions. This focus is particularly important in extreme environments, where mosses may find refuge from the prevailing macroclimatic conditions in microhabitats such as hypoliths.

## Introduction

Competitive exclusion and habitat selection theory posit that differential habitat selection may permit organisms of similar phenotypes to coexist [1–3]. This phenomenon can be observed in spatial and temporal differentiation, both of which are well-documented in plants [4].

Department of Biological Sciences Faculty-Student Achievement Fund to K.M.F.; the University of California, Berkeley, Department of Integrative Biology Graduate Research Fund to J.T.B.E.; the NIH-NIGMS Minority Biomedical Research Support-Research Initiative for Scientific Enhancement program (GM61331) to J.T.B.E.; and the NSF Dimensions of Biodiversity award (DEB-1638996 and DEB-1638956) to K.M.F. The funders had no role in study design, data collection and analysis, decision to publish, or preparation of the manuscript. Publication made possible in part by support from the Berkeley Research Impact Initiative (BRII) sponsored by the UC Berkeley Library.

**Competing interests:** The authors have declared that no competing interests exist.

Similarly, soil microorganisms exhibit habitat differentiation and resource partitioning, promoting species coexistence due to spatial heterogeneity [5]. Cryptobiosis, or dormancy, is common among soil microorganisms and can be understood as a form of temporal habitat differentiation, where organisms "occur" at different times in the same space. Biological soil crusts (biocrusts), communities of bryophytes, lichens, fungi, cyanobacteria, and other microorganisms living on the surface of the soil in deserts and drylands, exhibit impressive cryptobiosis [6]. Temporal partitioning is a critical strategy for these organisms, which may be desiccated and dormant for several consecutive months of each year. While many mosses are found in cool, low light environments, several species are abundant in deserts and drylands as important members of these biocrust communities.

As poikilohydric organisms, mosses equilibrate rapidly to ambient water content. This means that in the desert, which often experiences low humidity and high potential evapotranspiration, mosses can lose virtually all of their cellular free water and still resume normal growth once rehydrated; a complex trait known as desiccation tolerance [7]. Some of the most desiccation tolerant plants are species in the genus *Syntrichia*, which represent dominant members of Mojave Desert biocrust communities. However, even within harsh macroclimates, desert mosses may find climate buffering and more temperate conditions in the microenvironments that they occupy. For example, mosses are effective dew collectors, an important water source in very arid climates [8].

Similarly, mosses occupy microhabitats that may experience dramatically different light environments than the macroenvironment might suggest. For instance, many desert mosses occur under the shade of larger vascular plants [9,10], where they can take advantage of variable light and brief sun flecks when hydrated. Still, mosses are often found in open, exposed spaces, experiencing intensity of sunlight far beyond their light saturation points [11]. Thus, during hot, dry summer months, exposed biocrust mosses experience intense solar radiation with no ability to actively repair damage caused by UV and excess photosynthetically active radiation (PAR). Furthermore, while dry, these mosses are unable to use any PAR for photosynthesis. While quiescent, though, desiccated mosses do have passive avoidance strategies. Most mosses exhibit leaf-curling when dry, a mechanism that may reduce direct sunlight on leaf lamina [12,13]. Many desert mosses also have translucent leaf cells at the tips of their leaves, some even extending into long, hyaline awns, possibly reducing solar absorbance by increasing reflectance [12–14]. Furthermore, some mosses accumulate pigments such as carotenoids, anthocyanins, and UV-absorbing compounds such as flavonoids that may act as passive sunscreens [15–19].

Hypoliths are organisms that live under and on the belowground surface of translucent and opaque stones (typically quartz) that are embedded in the soil surface [20]. While they can occur anywhere suitable substrate is available, they are common in drylands [21], the largest terrestrial biome. Hypoliths experience enhanced water availability relative to surrounding soil organisms due lower evaporation, higher relative humidity (RH), and capture of water via fog condensation [21]. Nonetheless, dryland hypolithic habitats are still colonized by poikilohydric organisms that must withstand extended periods without water [22]. Cyanobacteria are the most common and dominant organisms in hypolithic communities [21], particularly taxa from the genus *Chroococcidiopsis* [23].

Moss can be observed growing adjacent to hypolithic cyanobacteria-harboring quartz rocks [20] and hypolithic communities may even be a necessary successional step for moss growth in some ecosystems [24,25]. Though less frequently, mosses also occur in hypolithic habitats, especially in extreme environments [20,26–29]. For instance, a single patch of *Tortula inermis* was reported under a crystalline rock in Death Valley [30], and there are additional reports of temperate hypolithic mosses [31], including an endemic obligate hypolith from Kansas in the

Great Plains of the United States [32]. Overall, studies including hypolithic mosses are limited and there are even fewer that aim to characterize the hypolithic moss community within a local area. This work serves as an important addition to this understudied topic, which has the potential to extend understanding of habitat partitioning and drivers of moss species diversity in arid environments. The main objective of this study was to compare hypolithic and soil surface moss communities in a western Mojave Desert wash. Specifically, we aimed to (1) compare relevant physical characteristics to determine whether microclimatic conditions differ significantly between hypolithic and adjacent soil surface microhabitats, and (2) test whether species composition or growth characteristics differ in hypolithic and adjacent soil surface microhabitats.

## Methods

### Site description & microclimate monitoring

The Sheep Creek Wash Mojave Desert site was visited and sampled in June of 2014. The site is at 1900 m elevation at the west end of the Mojave Desert and the northern base of the San Gabriel Mountains near Wrightwood, CA (34˚22'33.85"N, 117˚36'34.59"W). The average high and low annual temperatures are 16.3 ˚C and 1.6 ˚C, respectively, with an average annual precipitation of 49.4 cm (2005–2009, Wrightwood Weather Station, NOAA National Climatic Data Center). Soil mosses in this rocky wash grow in a semi-continuous carpet (Fig 1A and 1B).

In order to understand how the hypolithic microclimate compares to that of the surface, temperature and RH were measured with iButton hygrochrons (Maxim Integrated, San Jose, CA, USA) from September 2019 to February 2020. One iButton was deployed under a quartz rock that had moss growing under it and the other on a nearby soil surface moss (within 1 m of the quartz rock). Data were recorded once every hour and were summarized to find the high and low temperatures and relative humidities of each day. Mean daily highs and lows from each microhabitat type (soil surface and hypolithic) were compared using the paired Student's T-test. The difference in temperature and RH between surface and hypolithic microhabitats was calculated for each hourly time point. Differences in RH were binned into 20% bins, which were then used to calculate the proportion of total time that the difference in RH between microhabitats was greater than 10%.

### Light transmittance

Average amount of 650 nm light transmittance through sampled quartz rocks was calculated with a Beer's Law equation for Mojave Desert quartz pebbles [33]. These estimates were tested empirically with Onset HOBO Pendant temperature & light data loggers (Onset Computer Corporation, Bourne, MA, USA) and two milky quartz rocks collected from the study site (approximately 10 mm and 25 mm thick at center) in a growth chamber and in outdoors in full sunlight.

### Sample collection

Restriction of the hypolithic moss community to quartz rocks was first tested by pairing inspection under quartz rocks with inspection under non-quartz rocks of similar size within a 2 m x 2 m quadrat. To compare hypolithic and soil surface moss communities, samples were collected in approximately 0.5 cm clumps under and on the soil surface immediately adjacent to each quartz rock in a randomly selected 1 m x 1 m quadrat. Collection continued with sampling one quadrat every 3 m along two 15 m linear north-south (N-S) transects, 6 m apart

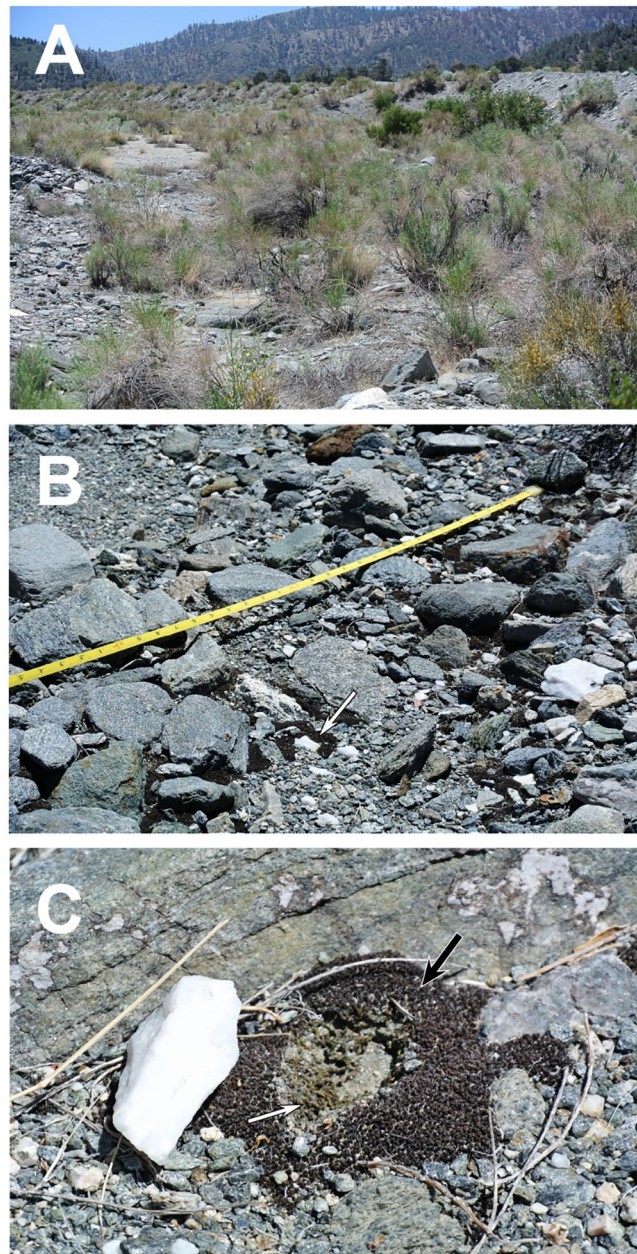

**Fig 1. Field site and habitat of soil surface and hypolithic mosses at Sheep Creek Wash, Mojave Desert, CA.** (A) Vegetation and environment in the study site. (B) Mosses growing in a rocky, semi-continuous carpet. Moss growing on the soil surface near a milky quartz rock (indicated with arrow). (C) Sampling sites for hypolithic (white arrow) and adjacent soil surface (black arrow) microhabitats.

from one another, for a total of 8 quadrats, 18 quartz rocks, and 53 moss samples. At the time of collection, quartz approximate thickness at center was measured to the nearest millimeter. Each sample was stored air-dried in a plastic box for subsequent species identification and shoot measurements. Samples were collected under a USDA US Forest Service permit to K. Fisher.

## Species composition

Shoots from each field collection were dissected and observed under dissecting and compound microscopes in both desiccated and hydrated states to identify to species using characteristic shoot morphology and leaf cross-sections [34–36]. Statistical difference between relative abundance of mosses in surface and hypolithic positions was tested with a 3 × 2 contingency table and Fisher's exact test [37].

## Shoot length & leaf density

Each of 349 *Syntrichia caninervis* shoots ($n_{HYP}$ = 50, $n_{SUR}$ = 299) was placed under a dissecting microscope to be measured digitally using a calibrated Motic microscope and software (Motic, Hong Kong, China). Only the length of shoot containing living tissue was measured. The boundaries of living tissue were determined by identifying leaves that had chlorophyllose tissue or other uniform pigmentation and that remained relatively closed when dry. Dead tissue, on the other hand, comprised open, damaged leaves with faded or blotchy pigmentation. Shoot lengths were first tested for normality with a Shapiro test [38] and then compared with a Wilcoxon test [39].

Leaf density was approximated on a subset of shoots. One shoot per remaining moss sample was selected at random and dead tissue was removed as above. Shoots were rehydrated and leaves were carefully removed and counted. Stem lengths were measured, and leaf density was calculated as number of leaves divided by stem length. Leaf density data were first tested for normality with a Shapiro test [38] and then compared with a Wilcoxon test [39].

## Results

### Microclimate

The mean daily high temperature on the soil surface was more than 2 ˚C warmer than in the hypolithic microhabitat under a quartz rock, while the mean daily low of the soil surface was almost 2 ˚C lower than the hypolithic space ($P < 0.0001$, Table 1). Over the microclimate monitoring period, soil surface temperatures were frequently warmer than the hypolithic microhabitat during the day and cooler at night (Fig 2). The quartz hypolithic microhabitat was slightly but consistently warmer during two periods of snow cover in November and December [40–43]. The mean daily low RH also differed between the soil surface and the hypolithic microhabitat. The mean daily low RH on the soil surface was 32.5% while in the hypolithic microhabitat under a quartz rock it was 62.5% ($P < 0.001$, Table 1). There was no significant difference in the mean daily high RH between the soil surface and the hypolithic microhabitat. During the first two months of microclimate monitoring, differences in RH were smaller in magnitude (within about 25%), with a general pattern of higher RH in the hypolithic microhabitat during the day and lower RH at night (Fig 3). However, from mid-November to end of

**Table 1. Microclimate in soil surface and hypolithic microhabitats from Sheep Creek Wash, Mojave Desert.**

|  | Surface | Hypolithic | *P*-value |
|---|---|---|---|
| Mean daily low temperature (˚C) | 5.65 ± 5.03 | 7.98 ± 5.35 | < 0.001 |
| Mean daily high temperature (˚C) | 21.6 ± 12.8 | 18.9 ± 11.6 | < 0.001 |
| Mean daily low relative humidity (%) | 32.5 ± 31.2 | 63.9 ± 39.5 | < 0.001 |
| Mean daily high relative humidity (%) | 71.7 ± 27.5 | 71.4 ± 34.9 | ns |

Paired Student's T-test, n = 158. ns = not significant.

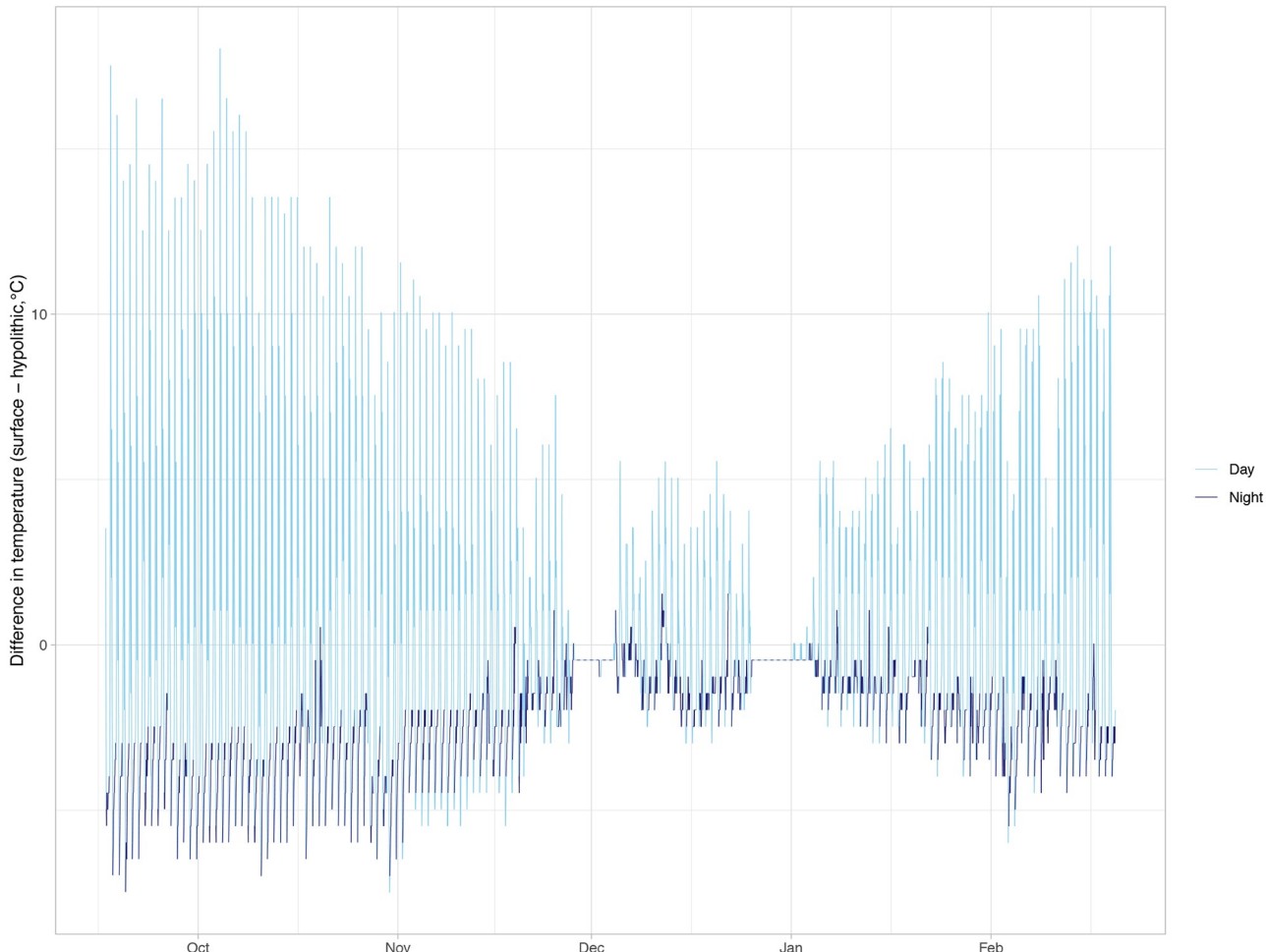

**Fig 2. Difference in temperature (°C) between soil surface and hypolithic microhabitats in Sheep Creek Wash over five months.** The difference in temperature between Sheep Creek Wash soil surface and quartz hypolithic microhabitats measured hourly from September 2019 to February 2020. Temperature difference is calculated as surface temperature—hypolithic temperature. Light blue line indicates "day" hours, from 6 am– 6 pm PDT, while dark blue line indicates "night" (6 pm– 6 am PDT).

February, RH was almost always higher in the hypolithic microhabitat, even at night. The times where RH was not higher under the quarts mostly correspond to two snow-covered periods [40–43] in which there was no difference in RH between the microhabitats. Over the monitoring period, 51.4% of the time RH was more than 10% higher in the hypolithic microhabitat compared to the soil surface. For 18.4% of the time, the hypolithic microhabitat was more than 10% lower in RH than on the soil surface, while 30.2% of the time the RH measurements in the two microhabitats were within 10% of each other.

## Quartz light transmittance

The average thickness of 18 quartz rocks that harbored hypolithic mosses in our study was approximately 26 ± 15 mm at the center, with rocks ranging from about 6 to 60 mm. According to the Beer's law Eq (1) of the line for transmission of 650 nm light as a function of Mojave Desert quartz thickness [33], only 0.065% of 650-nm light is transmitted through a 26 mm

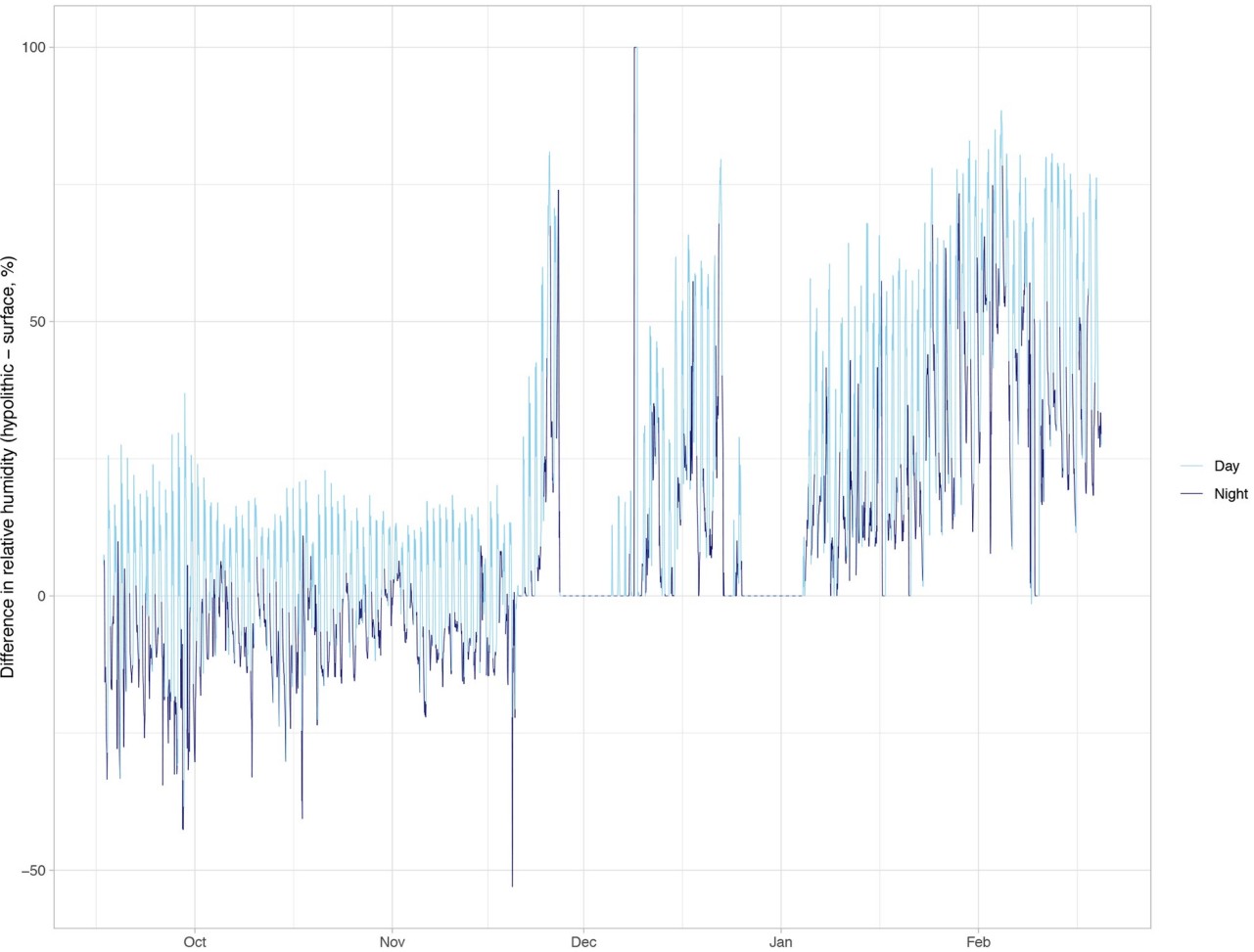

**Fig 3. Difference in percent relative humidity between soil surface and hypolithic microhabitats in Sheep Creek Wash over five months.** The difference in relative humidity (RH) between Sheep Creek Wash soil surface and quartz hypolithic microhabitats measured hourly from September 2019 to February 2020. RH difference is calculated as hypolithic RH—surface RH. Light blue line indicates "day" hours, from 6 am– 6 pm DST, while dark blue line indicates "night" (6 pm– 6 am DST).

Mojave Desert milky quartz rock.

$$\ln(\%T) = -0.261(t) + 3.961 \tag{1}$$

Average light transmittance for all 18 rocks that harbored hypolithic mosses in our study was found to be 1.2%, σ = 2.6% using Eq (1). According to our own measurements of light transmission through quartz collected from the study site, light intensity under a rock approximately 25 mm thick at center was 0.4% that of the exposed surface next to the rock. Under the 10 mm quartz rock, light was approximately 4% relative to surface intensity.

## Quartz restriction of hypoliths

Eight of nine quartz rocks in a 2 m × 2 m quadrat harbored some hypolithic moss, while none of the nine similarly sized non-quartz rocks in the same quadrat had mosses growing underneath.

**Table 2. Species composition and microhabitat contingency table.** Occurrences of *Syntrichia caninervis*, *Tortula inermis*, and *Bryum argenteum* under milky quartz rocks and on adjacent soil surface.

|  | *S. caninervis* | *T. inermis* | *B. argenteum* |
|---|---|---|---|
| Hypolithic | 12 | 12 | 0 |
| Surface | 24 | 3 | 2 |

Fisher's exact test; *P* = 0.003.

## Moss community composition

Of the 53 hypolithic and surface samples, 36 (68%) were *S. caninervis*, 15 (28%) were *T. inermis*, and 2 (4%) were *Bryum argenteum*. *Tortula inermis* was significantly more likely to be found in hypolithic microenvironments, while *S. caninervis* was more abundant on adjacent soil surfaces (Table 2, *P* = 0.003). Specifically, 24/36 (67%) of *S. caninervis* samples were found in soil surface positions adjacent to quartz rocks, and 12/15 (80%) of *T. inermis* were in hypolithic microhabitats (Fig 4A and 4B). Both samples of *B. argenteum*, a cosmopolitan weedy moss species, were found in soil surface microhabitats.

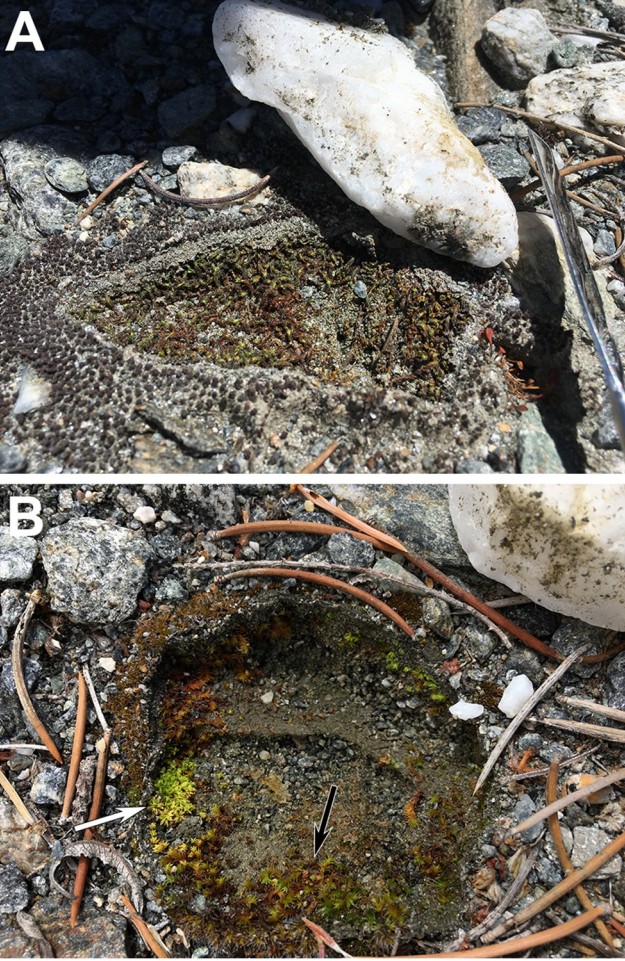

**Fig 4. Moss species from soil surface and hypolithic microhabitats in Mojave Desert Sheep Creek Wash.** (A) *Syntrichia caninervis* growing in both soil surface and milky quartz hypolithic microhabitats. (B) *Tortula inermis* (white arrow) and *S. caninervis* (black arrow) growing in a milky quartz hypolithic microhabitat.

### Habitat-dependent shoot length & leaf density

Hypolithic *S. caninervis* shoots were longer than soil surface shoots ($P < 0.0001$, Fig 5). The length of living shoots from the soil surface was 1.21 mm on average, while hypolithic shoots averaged 1.97, 62% longer than those collected from soil surface habitats. Hypolithic *S. caninervis* shoots had a lower leaf density than those from the soil surface ($P = 0.0125$, Fig 6). Shoots from hypolithic microhabitats had 16.5 leaves/mm on average, while soil surface shoots had 28.7 leaves/mm.

## Discussion

In this high elevation western Mojave Desert site, the living shoot tissue of *S. caninervis* was longer when growing in hypolithic microhabitats compared to the soil surface (Fig 3) and had lower leaf density (Fig 4), perhaps due to lower light [44]. Hypolithic mosses experience much lower light intensity than mosses on the soil surface, less than 4% of surface light intensity. Previous studies have found an average range of 50–99% of ambient PAR intensity reaching hypolithic spaces [25,28,45,46]. At the low end, Beer's Law extrapolation from integrating sphere measurements of light transmittance through Mojave Desert quartz pebbles finds an average of just 1.18% light transmittance for all 18 quartz rocks that harbored hypolithic mosses in this study [33]. On the other hand, lower light intensity could also be a benefit, even to photosynthetic organisms such as mosses, by way of reduction of photobleaching and energy burden to dissipate excess light [47]. Light transmission through Mojave milky quartz is relatively constant across the visible region of the electromagnetic spectrum but increases slightly from 390 nm wavelengths to 1090 nm wavelengths (approximately the upper limit of ultraviolet light to the upper limit of infrared light). This suggests the possibility that not only is the hypolithic space a refuge from overall high light intensity but that hypoliths also experience a smaller proportion of damaging UV light relative to photosynthetically active radiation [47]. In fact, there is evidence that nearly all UV-A and UV-B radiation is filtered out before reaching hypolithic communities [25,28]. *Syntrichia caninervis* develops a dark brown or black coloration in natural environments, a phenotypically plastic trait that does not occur in low-light laboratory conditions and may represent a UV sunscreen. This pigmentation was reduced or absent in quartz hypolithic microhabitats, possibly caused by the drastically different light environment. Interestingly, UV-B tolerance in mosses seems to correlate with desiccation tolerance [48] and *S. caninervis*, being one of the most desiccation-tolerant plants known, may also be expected to have high UV tolerance, too. Indeed, a close relative of this species, the also highly desiccation-tolerant *S. ruralis*, is not damaged by UV-B radiation, at least based on chlorophyll fluorescence [48].

Hypolithic *S. caninervis* plants may also be growing more, and thus have longer shoots, due to increased moisture retention resulting in an extended growing season relative to the adjacent soil surface [21] (Table 1). In this study, the area under quartz rocks was moist to the touch and the mosses were hydrated two weeks post-rain (assessed May 11, 2014; 0.51 mm rain at nearby Palmdale Airport on April 26, 2014; month-to-date rain: 10.9 mm). Microclimate monitoring found the soil surface to have a mean daily low RH much lower than in the adjacent hypolithic microhabitat (Table 1). Furthermore, we saw strong seasonal effects in RH differences between the microhabitats. In the warmer months, differences were smaller in magnitude and varied diurnally, with daytime having higher RH under quartz and night having higher RH on the soil surface. However, in winter the quartz hypolithic microhabitat almost always had higher RH than the soil surface. These data suggest the hypolithic spaces may act as a buffer to desiccation due to reduced evapotranspiration [27,49]. In contrast, desert soil surface mosses may desiccate within a day after a rainfall [50,51], even in as few as three

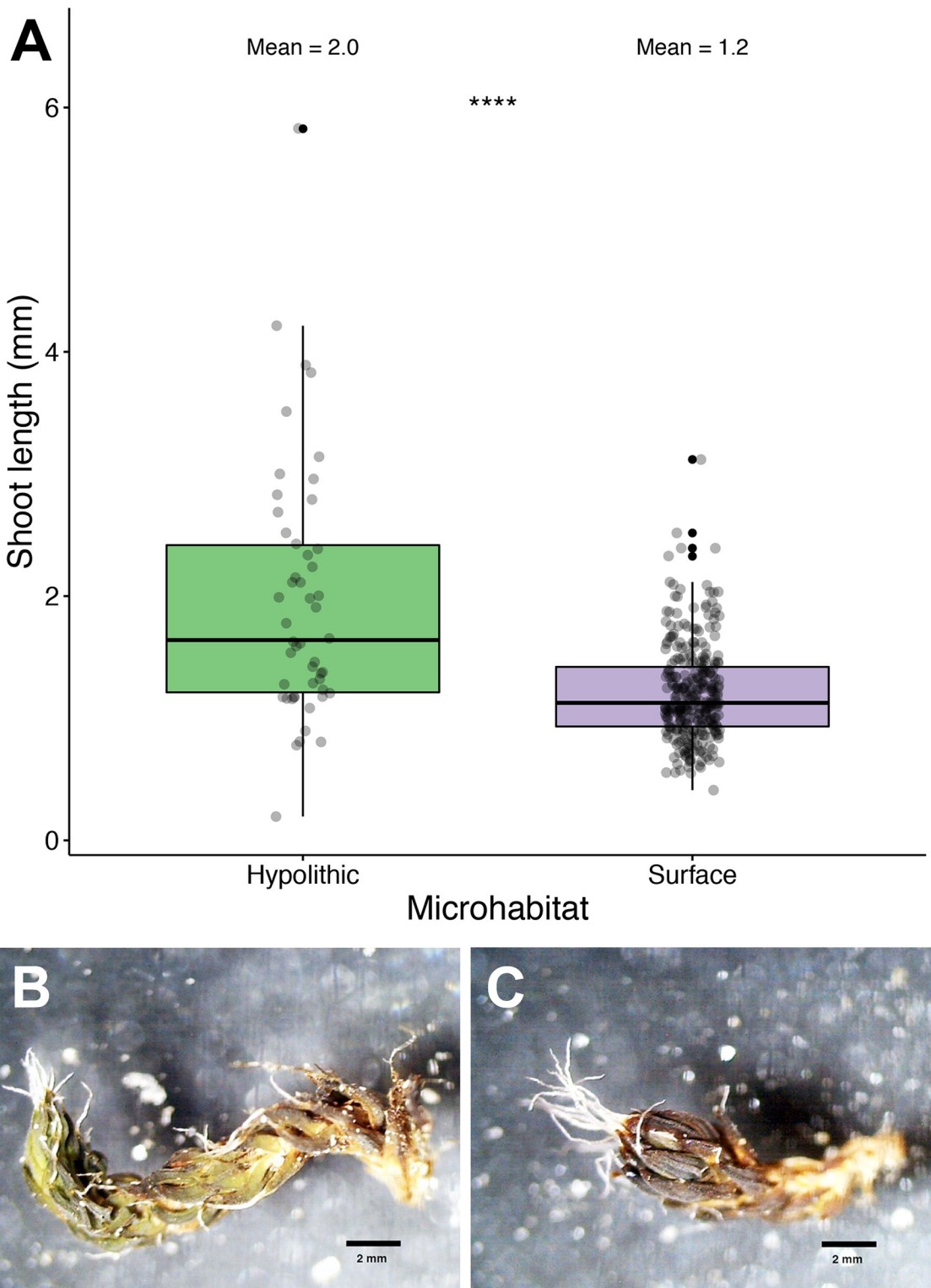

**Fig 5. Differential shoot length in *Syntrichia caninervis* shoots from hypolithic and soil surface microhabitats.** (A) Box plot of hypolithic and soil surface *S. caninervis* shoot length. **** Wilcoxon test, $P < 0.0001$. Mean$_{HYP}$ = 2.0 mm, mean$_{SUR}$ = 1.2 mm; n$_{HYP}$ = 50, n$_{SUR}$ = 299. (B) An *S. caninervis* shoot from a soil surface microhabitat. (C) An *S. caninervis* shoot from a hypolithic microhabitat.

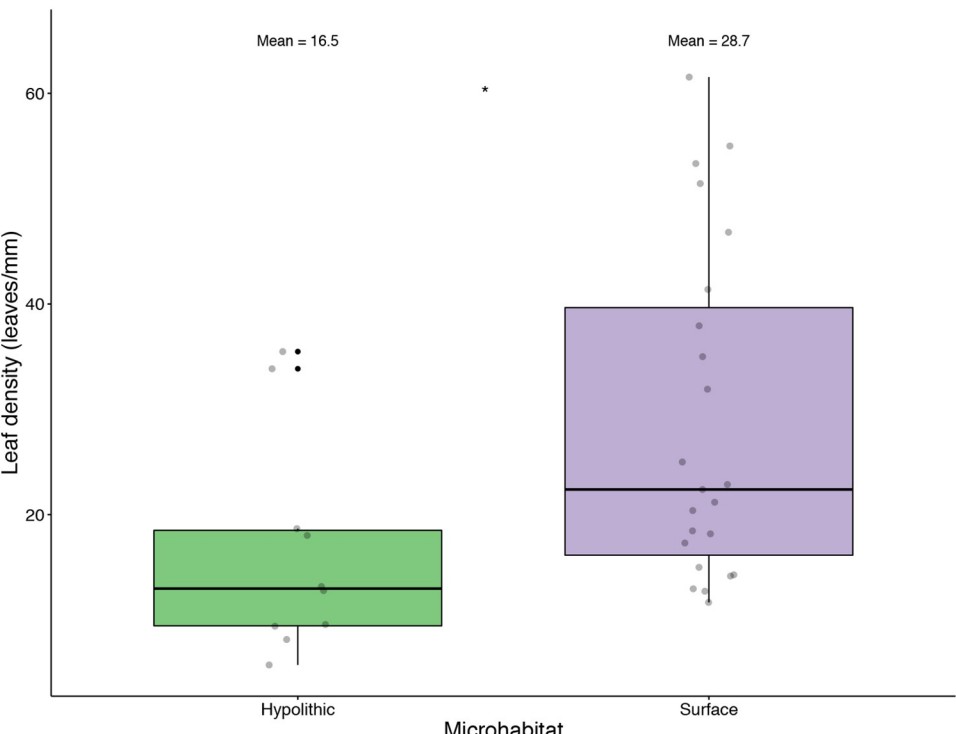

**Fig 6. Differential leaf density in *Syntrichia caninervis* shoots from hypolithic and soil surface microhabitats.**
Box plot of hypolithic and soil surface *S. caninervis* leaf density. * Wilcoxon test, *P* = 0.0125. Mean$_{HYP}$ = 16.5 leaves/mm, mean$_{SUR}$ = 28.7 leaves/mm; n$_{HYP}$ = 10, n$_{SUR}$ = 23.

hours [52]. The longest reported hydroperiod (time of complete hydration) for a Mojave Desert moss is 17 days, though most range between 1 and 4 days [52]. We found that nearly half of the monitoring period, the RH of hypolithic microhabitat was more than 10% higher than the soil surface, suggesting hypolithic mosses may be able take advantage of longer hydroperiods with more favorable RH. Indeed, hypolithic mosses receive very little light and would presumably need longer hydroperiods in order to take advantage of it. Longer hydroperiods not only allow mosses to remain photosynthetically active for a longer period of time, but overly brief periods of water availability may actually cause damage to desert mosses in the form of respiratory carbon deficit [53]. Similarly, soil moisture is higher under quartz in Antarctica [54], due in part to the tendency of meltwater draining around the edges of rocks and exposure of surface soil to wind drying [26].

Hypolithic microhabitats may also be providing refuge from extreme temperature fluctuations that are common in the Mojave Desert. Previous studies have reported hypolithic spaces to be up to 10 ˚C warmer than ambient air temperatures [33,54], even preventing freezing in winter. However, in at least one case, lower temperatures were reported under quartz rocks [26]. These apparently conflicting results may be due to thermal inertia of the rocks causing a lag in temperature changes [26]. In other words, after heating over the course of a day, quartz will cool off more slowly than the air, potentially preventing freezing in hypolithic spaces. Correspondingly, hypolithic spaces experience a slower rate of heating, potentially resulting in lower temperatures under rocks relative to adjacent surface or ambient air temperatures as temperatures rise. This thermal buffering results in less daily temperature variation in hypolithic habitats than in surrounding surface habitats, a phenomenon also seen in hypolithic microbial systems [27,46,55], which may also facilitate moss shoot growth under quartz.

Microclimate monitoring in this study supports this hypothesis. As seen in Table 1, mean daily high temperatures were lower in hypolithic microhabitats while mean daily low temperatures were higher in hypolithic microhabitats. Furthermore, we found a strong diurnal effect on temperature differences between the two microhabitats. During the day, temperatures were higher on the soil surface but at night, it was frequently warmer under the quartz, further suggesting quartz provides buffering from temperature extremes.

Because they lack roots, mosses rely upon external deposition and subsequent absorption of essential nutrients such as nitrogen and phosphorus [56]. In hypolithic spaces, mosses might experience limited access to nutrients typically acquired via atmospheric deposition, especially in ecosystems with low soil fertility like deserts. In both cold and hot deserts, hypoliths typically harbor a suite of cyanobacteria, but these taxa lack significant nitrogen fixation capacity [57,58]. When diazotrophic activity is present in hypoliths, it is generally accomplished by Proteobacteria [57]. Thus, despite the negative relationship between moss presence and cyanobacterial abundance in hypoliths [59], mosses growing in hypolithic niches could potentially acquire nitrogen from non-cyanobacterial diazotrophs.

Hypolithic moss species composition was distinct from soil surface species composition, with a higher prevalence of *T. inermis*. *Tortula inermis* typically occurs at lower elevations of the Mojave Desert than *S. caninervis*, which suggests that this species is adapted to hotter and drier conditions than *S. caninervis* [50]. In our study, *T. inermis* was much more likely to occupy protected hypolithic spaces than exposed surface conditions (Table 2). This finding, while initially counterintuitive, may highlight the importance of interactions between both temperature stress and moisture availability in controlling the distribution of this species. Dryland mosses are photosynthetically efficient at low light levels, which tend to prevail in winter months when populations are hydrated [60] and overcast conditions are ideal for growth [61]. At the elevation of our study site, these periods when mosses are metabolically active are also accompanied by extreme low temperatures and significant snowfall events. Two such events occurred during our microenvironmental monitoring (Nov. 27, 2019 for 8 days and Dec. 25, 2019 for 10 days). These periods when snow covered the soil surface are evidenced by a constant difference in the temperature and RH readings from hypoliths and the soil surface, with hypoliths maintaining the same RH and a slightly warmer temperature than the surface (Figs 2 and 3). Thus, at our study site, *T. inermis*, which is typically found at lower (i.e., warmer) elevations, may benefit from the thermal protection that hypolithic spaces provide during the growing season, and its prevalence in hypoliths may reflect lower levels of cold stress tolerance compared to *S. caninervis*. Although specific composition differed in the soil surface and quartz hypolithic microhabitats, *S. caninervis* was abundant in both. This pattern of distinct but overlapping communities in soil surface and hypolithic microhabitats has been found in other studies of hypolithic microbial systems [55,62]. *Syntrichia caninervis* grows as a semi-continuous carpet in this Mojave Desert site and frequently occurs in fully exposed microsites, as well as under the shade of shrubs [63,64]. This suggests that *S. caninervis* is perhaps more tolerant and physiologically plastic while *T. inermis* is may be restricted to hypolithic microhabitats in this high elevation site at the limits of its niche.

This study demonstrates that the desert hypolithic microenvironment provides conditions that support a different moss species composition and different growth patterns than the prevailing surface conditions. Our findings parallel those of prior work on microbial hypoliths that has also shown a community composition distinct from surrounding soils in terms of taxonomic abundance, but filtered from the regional pool of soil taxa by conditions unique to the hypolithic niche [24,62]. Furthermore, this work expands upon our understanding of habitat partitioning and drivers of moss species diversity in desert environments. Our data suggest that in the western high elevation Mojave Desert, lower light, thermal buffering, and longer

hydroperiods contribute to a higher representation of *Tortula inermis* and increased growth for the dominant moss *Syntrichia caninervis* in hypolithic microhabitats than on the soil surface. Although the hypolithic moss habitat is relatively understudied, the concept of microenvironment for mosses is not a new one [10,65–67]. Even desert soil surface mosses tend to occupy specific microenvironments, such as in the shade of a shrub or on the north side of a boulder, where prevailing temperature and moisture conditions are buffered from those of the overall macroenvironment. Yet, the scale at which mosses experience their environment is probably as proportionate to their body size as the macroclimate is to macroorganisms. Landscape features such as mountain ranges are broadly appreciated for their influence on physical environmental conditions and associated species distributions; here we have shown that for smaller organisms, analogous microenvironmental features are not trivial, and can likewise influence community composition. In sum, this study reinforces the need to consider microenvironmental conditions and their variation in the characterization, prediction, and conservation of bryophyte communities.

## Supporting information

**S1 Video.**
(MOV)

## Acknowledgments

We thank the students from Cal State LA and Berkeley High School who have contributed to this project.

## Author Contributions

**Conceptualization:** Jenna T. B. Ekwealor, Kirsten M. Fisher.

**Data curation:** Jenna T. B. Ekwealor.

**Formal analysis:** Jenna T. B. Ekwealor.

**Investigation:** Jenna T. B. Ekwealor, Kirsten M. Fisher.

**Methodology:** Jenna T. B. Ekwealor, Kirsten M. Fisher.

**Supervision:** Kirsten M. Fisher.

**Visualization:** Jenna T. B. Ekwealor.

**Writing – original draft:** Jenna T. B. Ekwealor, Kirsten M. Fisher.

**Writing – review & editing:** Jenna T. B. Ekwealor, Kirsten M. Fisher.

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
