## [Decision Letter · Decision Letter 0]

29 Apr 2020

PONE-D-20-08918

Life under quartz: Hypolithic mosses in the Mojave Desert

PLOS ONE

Dear Ms. Ekwealor,

Thank you for submitting your manuscript to PLOS ONE. After careful consideration, we feel that it has merit but does not fully meet PLOS ONE’s publication criteria as it currently stands. Therefore, we invite you to submit a revised version of the manuscript that addresses the points raised during the review process.

We would appreciate receiving your revised manuscript by Jun 12 2020 11:59PM. To enhance the reproducibility of your results, we recommend that if applicable you deposit your laboratory protocols in protocols.io, where a protocol can be assigned its own identifier (DOI) such that it can be cited independently in the future. For instructions see: http://journals.plos.org/plosone/s/submission-guidelines#loc-laboratory-protocols

We look forward to receiving your revised manuscript.

Kind regards,

Matthew Germino, Ph.D.

Academic Editor

PLOS ONE

Additional Editor Comments:

We were able to obtain two thoughtful and useful reviews for your paper. Both reviewers and I think you have a nice contribution to the literature, and all of the reviewer comments should prove valuable in some minor touch ups to the MS. Please note that PLoS One does little to no editing for style, grammar, and typos...as reviewer 1 notes, the mixing of citation styles should be addressed. Congratulations on a nice study and paper.

Journal Requirements:

3. In your Data Availability statement, you have specified a link for the minimal data set however the link address given brings an error message; the data cannot be found.

PLOS defines a study's minimal data set as the underlying data used to reach the conclusions drawn in the manuscript and any additional data required to replicate the reported study findings in their entirety. All PLOS journals require that the minimal data set be made fully available. For more information about our data policy, please see http://journals.plos.org/plosone/s/data-availability.

Reviewers' comments:

Reviewer's Responses to Questions

**Comments to the Author**

1. Is the manuscript technically sound, and do the data support the conclusions?

Reviewer #1: Yes

Reviewer #2: Yes

2. Has the statistical analysis been performed appropriately and rigorously? 

Reviewer #1: Yes

Reviewer #2: Yes

3. Have the authors made all data underlying the findings in their manuscript fully available?

Reviewer #1: Yes

Reviewer #2: Yes

4. Is the manuscript presented in an intelligible fashion and written in standard English?

Reviewer #1: Yes

Reviewer #2: Yes

5. Review Comments to the Author

Reviewer #1: This is a nice little MS examining desert moss community structure and plant growth under quartz rocks, a subject of perennial interest, especially in dryland ecosystems. The current MS is well-written, addressing a straightforward reseach question and goal with a basic, but adequate, data set and analysis, and I have no problems with any of the conclusions the authors come to regarding the advantages of the hypolithic environment for supporting moss persistence (and applaud them for expanding the denizens of such habitats to include multi-cellular plants) . I do have some changes that I think the authors could make to strengthen what is already a pretty robust effort:

1) There is enough information in the literature to formulate a true testable hypothesis regarding community composition and growth characteristics between inter- and under-rock micro-environments. For instance, one of the MS cited (Hamerlynck et al. 2002) showed that desiccation-tolerant mosses in shaded and exposed microenvironments in many ways "flipped" typical sun-plant and shade-plant attributes in vascular plants, and attributed this to the length of time out of the desiccated state. One could make a good case that, even with the dramatic attenuation of maximum PPFD intensity under quartz, enhanced moisture trapping and amelioration of temperature extremes such rocks provide could indeed favor much more prolonged periods of plant activity compared to the "boom-bust" on the surface.

2) I think referring to hypolithic mosses as "etiolated" in any way is unwarranted. Etiolation is an induced behavior of light seeking which combines excessive cellular elongation and is associated with a lack of pigment production and pigment degradation. If these mosses were in a microhabitat that resulted in etiolation, well, they just wouldn't be there long enough to establish in the first place. These are not etiolated plants, they are well-adjusted and fully adapted to the conditions they have established and persist in, and your structural data clearly support this assertion.

3) In light of #1 (above), mean relative humidity and temperature, though indicative of the broad differences between these habitats, are not what these plants are responding to, and you would be better served to fully present such differences. Rather, it is the differences in the cumulative period of favorable moisture and associated temperatures that are important. The authors have the detailed data, and can do more with it. A few suggestions: (a) generate probability density function for bins of humidity differences between surface and hypolithic microenvironments (a basic ecohydrological approach used frequently in inter-canopy vs under-canopy comparisons in dryland ecosystems - delve into Russell L. Scott's work (https://scholar.google.com/citations?user=3QI1SOMAAAAJ&hl=en), he uses them. He uses them a lot.) (b) quantify the average periods of time for bins of humidity and temperature differences between hypolithic and surface locations. The critical thing to establish is how much more frequent and how much longer conditions are favorable for hypolithic mosses, which is the critical feature that allows them to effectively utilize what are very low PPFD. This has been done for these mosses in more typical microhabitats, but not for this unique setting. There may be other creative ways to present data to reflect this, so ponder it a bit, and come up with a way to do so!

4) These mosses are not alone under there, as previous work by Schlesinger et al. (2004) show. Sharing space with blue-green algae and other photosynthetic prokaryotes, which are critical lynch-pins in dryland nutrient dynamics (especially nitrogen, and not just in the Antarctic, BTW), is likely going to help hypolithic mosses as well, especially in nutrient availability and subsequent photosynthetic pigment/enzyme production. Desiccation-tolerant mosses get pretty much everything via atmospheric deposition, which tends to be dry deposition in desert systems. Being under a rock is not going to be all that great a place to get much dry deposition (though it could be mobilized and accumulate following rain), but proximity to N-fixers over longer and more frequent humidity conditions would increase the overall ability of these mosses to get those limiting resources. Just a point to consider for your results, and future research directions.

5) Minor annoyed gruffing: keep your citation format in the text body consistent; there are several occasions where you shift from reference number to full citations (i.e. Whozits et al. 1883, rather than [1]).

6) From the "If This Were a Perfect World" files: if you at all retained the plant samples you used, getting an estimate of individual plant mass would be invaluable. Having plant mass alone would be of interest in and of itself (hello biomass!), but you could also estimate specific length (g dry mass m-1 of shoot length) of each plant, which, though not as informative as specific mass (g dry mass m-2 surface area), would provide information on total C-acquisition capacity between surface and hypolithic plants. If you don't, well, I'll be content to live with the study you did, not the one you didn't.

Reviewer #2: The manuscript, “Life under quartz: hypolithic mosses in the Mojave Desert” is concise & well-written. The study’s main objective was to identify the taxonomic diversity of mosses beneath hypolithic quartz rocks and to compare it to that of adjacent soil surface communities. The authors rightly point out that little is known about the unique moss habitat compared to adjacent moss living freely on the bare soil surface. These microbial communities are integral to an understanding of ecosystem services in desert environments, and an understanding of their biodiversity is also critical as a baseline in observing changes to the Mojave Desert as climate change progresses.

The study’s main findings found 1) data from microclimate loggers support the hypothesis that quartz provides microclimate buffering compared to the soil surface; 2) environmental differences enable niche partitioning, with T. inermis much more likely to occupy low light hypolithic spaces and S. caninervis more prevalent in hotter and drier exposed surface conditions, enhancing alpha diversity; 3) the hypolithic environment is buffered compared to bare soil surface microhabitat, particularly with respect to a more clement moisture regime that does not subject the community to as severe desiccation as the bare surface; 4) none of the similarly-sized non-quartz rocks supported hypolithic moss communities and 5) hypolithic moss species composition is distinct from soil surface species composition- i.e., hypolithic habitat supports both different moss species and different growth patterns than prevailing surface conditions. These findings will be of interest to the general microbial and specific desert ecology fields, along with those studying Mars analog extreme environments.

The results and discussion sections were clear and interesting, with some fascinating points. I really enjoyed reading the paper- well done!

I recommend publication after addressing the following minor comments.

1. Fig 1 C. In this picture you see the green hypolithic moss below the quartz rock and the dark black moss community on the periphery surrounding the quartz pebble. Are they different species or the same (I am assuming only one S. caninervis given the figure legend) species with different pigments? Is the dark black in the picture the moss you are referring to in the text as “on the bare soil surface”? In other words, I am a bit confused as to what exactly you are comparing, ie, did you put the ibutton in the green moss pictured here under the quartz and then under? the black moss directly adjacent shown here? Or was the moss you were comparing to the hypolithic one located some distance away on a bare surface nearby? Please clarify.

2. In light of #1, for your conclusions (line 242-244), T. inermis much more likely to occupy low light hypolithic spaces and S. caninervis more prevalent in hotter and drier exposed surface conditions. But, if there is also the same species in both the hypolithic and adjacent soil surface as well as in either/or it would also seem to me that S. caninervis is perhaps more hardy and adaptable (generalist?) to either the hypolithic or the bare surface microhabitat—ie it doesn’t really care which microhabitat it is in (high elevation, low elevation, open soil or hypolithic habitat) whereas the T. inermis is more finicky and/or less adaptable (as you reference in terms of its restriction to cooler, moister conditions). This also means that S. caninervis is less sensitive to high UV/light conditions as well, since it is pretty much in most available micro-habitats in your study area compared to the other species?

3. I would suggest that with your results in Table 1 it is (clearly and mostly) the higher % RH in terms of the low that controls this niche partitioning situation. Would you agree and if so, please also emphasize this. Very interesting.

4. Line 79-80. No active repair but obviously avoidance strategies including pigmentation—if so, please add a line or two on this in that same paragraph. Line 262-264—they are also clearly using pigmentation so please also mention that here as well. If you have any info on the differences in pigmentation, please include it.

5. For figure 1 can you please also add a picture of the other two types of moss in your study for those of us not very familiar with desert mosses.

6. Line 235-236: hypolithic moss species composition is distinct from soil surface species composition. But they also overlap as well, with both S. caninervis and T. inermis being found in both the hypolithic and the open surface as well. Please also mention this and that this has also been found for other hypoliths versus bare surface soil (e.g, Namib Desert, see Stomeo et al. 2013 and other more recent refs and also Antarctica).

7. The discussion is a bit light. Can you add a paragraph to put your findings in greater context for both mosses and for other hypolithic microbial systems and in comparison to BSC in the Mojave and perhaps with other deserts (e.g., Namib Desert hypoliths vs bare surface soil communities, Stomeo et al. 2013) such as Sonoran. You have done a good job mentioning other studies via the refs but I think this warrants a separate paragraph to elaborate upon. Also, one ref to consider including is Mogul et al. 2017. Please also extend the discussion a bit more to summarize on how your results expand our understanding of habitat partitioning (line 99 and 298).

8. Line 199-201: was there moss growing only on the surface in those quadrats— ie not associated with any quartz rocks and just by itself? If so, please elaborate on the implications of this, depending on if one or all three mosses are found on bare soil surface (or nearby under shade plants). This would point to the key advantage of the quartz being the increased moisture under the rocks if they are also out in open bare surface soil or if they are also only found under shade plants.

6. PLOS authors have the option to publish the peer review history of their article (what does this mean?). If published, this will include your full peer review and any attached files.

Reviewer #1: No

Reviewer #2: No

---

## [Author Response · Author response to Decision Letter 0]

3 Jun 2020

Reviewer #1: This is a nice little MS examining desert moss community structure and plant growth under quartz rocks, a subject of perennial interest, especially in dryland ecosystems. The current MS is well-written, addressing a straightforward reseach question and goal with a basic, but adequate, data set and analysis, and I have no problems with any of the conclusions the authors come to regarding the advantages of the hypolithic environment for supporting moss persistence (and applaud them for expanding the denizens of such habitats to include multi-cellular plants) . I do have some changes that I think the authors could make to strengthen what is already a pretty robust effort:

1) There is enough information in the literature to formulate a true testable hypothesis regarding community composition and growth characteristics between inter- and under-rock micro-environments. For instance, one of the MS cited (Hamerlynck et al. 2002) showed that desiccation-tolerant mosses in shaded and exposed microenvironments in many ways "flipped" typical sun-plant and shade-plant attributes in vascular plants, and attributed this to the length of time out of the desiccated state. One could make a good case that, even with the dramatic attenuation of maximum PPFD intensity under quartz, enhanced moisture trapping and amelioration of temperature extremes such rocks provide could indeed favor much more prolonged periods of plant activity compared to the "boom-bust" on the surface.

- We thank the reviewer for this comment. While we agree this would be interesting and insightful to compare to other systems, unfortunately we don’t believe it is possible within this study. Within bryology, and especially desert bryology, we are still learning basic ecology and natural history of many species. At present, the authors do not feel they have enough understanding of the ecology of T. inermis to make stronger predictions or conclusions than those we have. However, we have added more discussion about the possible niche of T. inermis and how that relates to this study (Lines 361-374).

2) I think referring to hypolithic mosses as "etiolated" in any way is unwarranted. Etiolation is an induced behavior of light seeking which combines excessive cellular elongation and is associated with a lack of pigment production and pigment degradation. If these mosses were in a microhabitat that resulted in etiolation, well, they just wouldn't be there long enough to establish in the first place. These are not etiolated plants, they are well-adjusted and fully adapted to the conditions they have established and persist in, and your structural data clearly support this assertion.

- Thank you for this insight. We have removed the lines about etiolation.

3) In light of #1 (above), mean relative humidity and temperature, though indicative of the broad differences between these habitats, are not what these plants are responding to, and you would be better served to fully present such differences. Rather, it is the differences in the cumulative period of favorable moisture and associated temperatures that are important. The authors have the detailed data, and can do more with it. A few suggestions: (a) generate probability density function for bins of humidity differences between surface and hypolithic microenvironments (a basic ecohydrological approach used frequently in inter-canopy vs under-canopy comparisons in dryland ecosystems - delve into Russell L. Scott's work (https://scholar.google.com/citations?user=3QI1SOMAAAAJ&hl=en), he uses them. He uses them a lot.) (b) quantify the average periods of time for bins of humidity and temperature differences between hypolithic and surface locations. The critical thing to establish is how much more frequent and how much longer conditions are favorable for hypolithic mosses, which is the critical feature that allows them to effectively utilize what are very low PPFD. This has been done for these mosses in more typical microhabitats, but not for this unique setting. There may be other creative ways to present data to reflect this, so ponder it a bit, and come up with a way to do so!

- Thanks for the clear and apt suggestions. We have created histograms of differences between the two habitats in temperature and relative humidity (RH) at every time point (once every hour) for the whole 5 months of monitoring. We used these histograms to summarize the amount of time that RH is more favorable (defined as >10% higher), less favorable (>10% lower), and not different (+-10%) in the two habitats. We have included these new methods in lines 137-140, and results are included in lines 188-191 and 196-205. We also created two new plots to visualize the differences in temperature and RH over the 5 months in Fig 2 and Fig 3. We believe these figures show striking patterns in the microclimate differences, on both a diurnal and seasonal scale. 

4) These mosses are not alone under there, as previous work by Schlesinger et al. (2004) show. Sharing space with blue-green algae and other photosynthetic prokaryotes, which are critical lynch-pins in dryland nutrient dynamics (especially nitrogen, and not just in the Antarctic, BTW), is likely going to help hypolithic mosses as well, especially in nutrient availability and subsequent photosynthetic pigment/enzyme production. Desiccation-tolerant mosses get pretty much everything via atmospheric deposition, which tends to be dry deposition in desert systems. Being under a rock is not going to be all that great a place to get much dry deposition (though it could be mobilized and accumulate following rain), but proximity to N-fixers over longer and more frequent humidity conditions would increase the overall ability of these mosses to get those limiting resources. Just a point to consider for your results, and future research directions.

- We appreciate this advice and have incorporated microbial community into the discussion (lines 348-356). 

5) Minor annoyed gruffing: keep your citation format in the text body consistent; there are several occasions where you shift from reference number to full citations (i.e. Whozits et al. 1883, rather than [1]).

- We have corrected these, thank you. Sincere apologies for the annoyance!

6) From the "If This Were a Perfect World" files: if you at all retained the plant samples you used, getting an estimate of individual plant mass would be invaluable. Having plant mass alone would be of interest in and of itself (hello biomass!), but you could also estimate specific length (g dry mass m-1 of shoot length) of each plant, which, though not as informative as specific mass (g dry mass m-2 surface area), would provide information on total C-acquisition capacity between surface and hypolithic plants. If you don't, well, I'll be content to live with the study you did, not the one you didn't.

- This is an interesting suggestion and we appreciate it. Unfortunately between tissue limitation (we sampled very conservatively the first time and several samples have no un-dissected stems left), and ongoing shelter-in-place orders in the area preventing us from even reaching the samples, these additional measurements will not be possible for this study at this time. However we do thank the reviewer for the suggestion as it may help us design better future studies. 

Reviewer #2: The manuscript, “Life under quartz: hypolithic mosses in the Mojave Desert” is concise & well-written. The study’s main objective was to identify the taxonomic diversity of mosses beneath hypolithic quartz rocks and to compare it to that of adjacent soil surface communities. The authors rightly point out that little is known about the unique moss habitat compared to adjacent moss living freely on the bare soil surface. These microbial communities are integral to an understanding of ecosystem services in desert environments, and an understanding of their biodiversity is also critical as a baseline in observing changes to the Mojave Desert as climate change progresses.

The study’s main findings found 1) data from microclimate loggers support the hypothesis that quartz provides microclimate buffering compared to the soil surface; 2) environmental differences enable niche partitioning, with T. inermis much more likely to occupy low light hypolithic spaces and S. caninervis more prevalent in hotter and drier exposed surface conditions, enhancing alpha diversity; 3) the hypolithic environment is buffered compared to bare soil surface microhabitat, particularly with respect to a more clement moisture regime that does not subject the community to as severe desiccation as the bare surface; 4) none of the similarly-sized non-quartz rocks supported hypolithic moss communities and 5) hypolithic moss species composition is distinct from soil surface species composition- i.e., hypolithic habitat supports both different moss species and different growth patterns than prevailing surface conditions. These findings will be of interest to the general microbial and specific desert ecology fields, along with those studying Mars analog extreme environments.

The results and discussion sections were clear and interesting, with some fascinating points. I really enjoyed reading the paper- well done!

I recommend publication after addressing the following minor comments.

1. Fig 1 C. In this picture you see the green hypolithic moss below the quartz rock and the dark black moss community on the periphery surrounding the quartz pebble. Are they different species or the same (I am assuming only one S. caninervis given the figure legend) species with different pigments? Is the dark black in the picture the moss you are referring to in the text as “on the bare soil surface”? In other words, I am a bit confused as to what exactly you are comparing, ie, did you put the ibutton in the green moss pictured here under the quartz and then under? the black moss directly adjacent shown here? Or was the moss you were comparing to the hypolithic one located some distance away on a bare surface nearby? Please clarify.

- Thank you for pointing out the ambiguities in how we presented these methods and this figure. We have added details to the methods (lines 153-155), added arrows to figures 1B and 1C, and have added more details to figure 1C legend. We have also added an additional figure, Figure 4, to more clearly demonstrate growth forms of these mosses in the different microhabitats. Finally, we added microscopic images of S. caninervis stems as an examplar of shoot length and pigmentation differences. 

2. In light of #1, for your conclusions (line 242-244), T. inermis much more likely to occupy low light hypolithic spaces and S. caninervis more prevalent in hotter and drier exposed surface conditions. But, if there is also the same species in both the hypolithic and adjacent soil surface as well as in either/or it would also seem to me that S. caninervis is perhaps more hardy and adaptable (generalist?) to either the hypolithic or the bare surface microhabitat—ie it doesn’t really care which microhabitat it is in (high elevation, low elevation, open soil or hypolithic habitat) whereas the T. inermis is more finicky and/or less adaptable (as you reference in terms of its restriction to cooler, moister conditions). This also means that S. caninervis is less sensitive to high UV/light conditions as well, since it is pretty much in most available micro-habitats in your study area compared to the other species?

- Yes, thanks for these questions and comments. We have changed the discussion to incorporate these points (lines 297-305 and 374-382). 

3. I would suggest that with your results in Table 1 it is (clearly and mostly) the higher % RH in terms of the low that controls this niche partitioning situation. Would you agree and if so, please also emphasize this. Very interesting.

- We have included new RH analyses (lines, Figure, lines 137-140 and 196-205), and have added emphasis of the importance of RH differences in the discussion (lines 312-324). 

4. Line 79-80. No active repair but obviously avoidance strategies including pigmentation—if so, please add a line or two on this in that same paragraph. Line 262-264—they are also clearly using pigmentation so please also mention that here as well. If you have any info on the differences in pigmentation, please include it.

- Thank you for the suggestion. We have included discussion of passive avoidance strategies such as leaf curling and pigmentation possibilities in the introduction and in the discussion (lines 312-324). 

5. For figure 1 can you please also add a picture of the other two types of moss in your study for those of us not very familiar with desert mosses.

- We have added photos of the different species of hypolithic mosses to Figures 4 and 5. 

6. Line 235-236: hypolithic moss species composition is distinct from soil surface species composition. But they also overlap as well, with both S. caninervis and T. inermis being found in both the hypolithic and the open surface as well. Please also mention this and that this has also been found for other hypoliths versus bare surface soil (e.g, Namib Desert, see Stomeo et al. 2013 and other more recent refs and also Antarctica).

- We have added this point to the discussion (lines 374-382 and 385-388). 

7. The discussion is a bit light. Can you add a paragraph to put your findings in greater context for both mosses and for other hypolithic microbial systems and in comparison to BSC in the Mojave and perhaps with other deserts (e.g., Namib Desert hypoliths vs bare surface soil communities, Stomeo et al. 2013) such as Sonoran. You have done a good job mentioning other studies via the refs but I think this warrants a separate paragraph to elaborate upon. Also, one ref to consider including is Mogul et al. 2017. Please also extend the discussion a bit more to summarize on how your results expand our understanding of habitat partitioning (line 99 and 298).

- We have added more discussion about hypolithic microbial systems (lines 348-356 and 385-388) and habitat partitioning (lines 380-382). 

8. Line 199-201: was there moss growing only on the surface in those quadrats— ie not associated with any quartz rocks and just by itself? If so, please elaborate on the implications of this, depending on if one or all three mosses are found on bare soil surface (or nearby under shade plants). This would point to the key advantage of the quartz being the increased moisture under the rocks if they are also out in open bare surface soil or if they are also only found under shade plants.

- Yes, mosses at this site grow in a semi-continuous carpet between shrubs and possibly not associated with any quartz rocks. Although our surface sampling for this study focused on collecting immediately adjacent to quartz rocks, previous studies at this site have found similar composition to our surface community composition (Baughman et al., 2017 and unpublished data). We have added more discussion on this point (lines 378-382), thank you.

---

## [Editor Report · Decision Letter 1]

25 Jun 2020

Life under quartz: Hypolithic mosses in the Mojave Desert

PONE-D-20-08918R1

Dear Dr. Ekwealor,

We’re pleased to inform you that your manuscript has been judged scientifically suitable for publication and will be formally accepted for publication once it meets all outstanding technical requirements.

Kind regards,

Matthew Germino, Ph.D.

Academic Editor

PLOS ONE

Additional Editor Comments (optional):

The revision looks great, except that the two new graphs done in excell have very small fonts and are not really formatted in a way that will work with standard typesetting. Note that PLoS One does not copyedit after acceptance.
---

## [Editor Report · Acceptance letter]

29 Jun 2020

PONE-D-20-08918R1 

Life under quartz: Hypolithic mosses in the Mojave Desert 

Dear Dr. Ekwealor:

I'm pleased to inform you that your manuscript has been deemed suitable for publication in PLOS ONE. Congratulations! Your manuscript is now with our production department. 

Kind regards, 

on behalf of

Dr. Matthew Germino 

Academic Editor

PLOS ONE